# Long-Duration Space Travel Support Must Consider Wider Influences to Conserve Microbiota Composition and Function

**DOI:** 10.3390/life12081163

**Published:** 2022-07-30

**Authors:** Kait F. Al, John A. Chmiel, Gerrit A. Stuivenberg, Gregor Reid, Jeremy P. Burton

**Affiliations:** 1Department of Microbiology and Immunology, University of Western Ontario, London, ON N6A 3K7, Canada; kal@uwo.ca (K.F.A.); jchmiel4@uwo.ca (J.A.C.); gstuiven@uwo.ca (G.A.S.); gregor@uwo.ca (G.R.); 2Department of Surgery, University of Western Ontario, London, ON N6A 4V2, Canada; 3Lawson Health Research Institute, London, ON N6A 4V2, Canada

**Keywords:** microbiome, gut microbiota, spaceflight, fermented food, probiotics, astronauts

## Abstract

The microbiota is important for immune modulation, nutrient acquisition, vitamin production, and other aspects for long-term human health. Isolated model organisms can lose microbial diversity over time and humans are likely the same. Decreasing microbial diversity and the subsequent loss of function may accelerate disease progression on Earth, and to an even greater degree in space. For this reason, maintaining a healthy microbiome during spaceflight has recently garnered consideration. Diet, lifestyle, and consumption of beneficial microbes can shape the microbiota, but the replenishment we attain from environmental exposure to microbes is important too. Probiotics, prebiotics, fermented foods, fecal microbiota transplantation (FMT), and other methods of microbiota modulation currently available may be of benefit for shorter trips, but may not be viable options to overcome the unique challenges faced in long-term space travel. Novel fermented food products with particular impact on gut health, immune modulation, and other space-targeted health outcomes are worthy of exploration. Further consideration of potential microbial replenishment to humans, including from environmental sources to maintain a healthy microbiome, may also be required.

## 1. Introduction

The advancement of space exploration has motivated investigation into the effects of spaceflight on human health. Since there has been growing recognition that the microbiome is instrumental in maintaining health, studies have begun to explore how the microbiome is shaped by spaceflight [1]. Conditions such as microgravity, cosmic radiation, and spaceflight can exert adverse changes on an astronaut’s microbiome, and individual bacterial physiology and virulence [2,3,4,5]. Unless actively maintained, over time an astronaut’s microbiota may be usurped by a “bloom” of predominant microbes that suppress the indigenous commensals. With an unknown potential for recovery and health implications, this could be problematic for space voyagers. Maintaining a healthy, diverse, and well-functioning microbiome will be a continuing challenge that may influence a space mission’s success. To this end, optimizing food consumed and other microbial exposures during spaceflight will be instrumental in ensuring that astronauts are able to live long and prosper (Figure 1).

A wealth of research goes into the creation of space food systems, particularly to optimize superior stability, nutrition, and palatability; however, the lack of microbial biomass may have been overlooked [6]. To date, the emphasis has been on maintaining the near sterility of a space station’s contents to reduce microbiological threats to the astronauts. However, diminished exposure to non-pathogenic environmental and dietary microbes over very long space travel may pose yet unknown health challenges. While space missions will carry a supply of medical interventions, the concept of targeted microbiome modulation is still in its nascent stages. Nurturing or replacing a healthy microbiome will need to be considered as space travel becomes increasingly widespread and as the voyages become longer. This article will explore some of the threats posed to an astronaut’s microbiome and potential approaches to mitigate them.

## 2. Spaceflight Is Hostile toward Humans and Microbes Alike

Spaceflight conditions are extremely hostile to the human body. The prolonged exposure to radiation, microgravity, acceleration forces, isolation, and emotional stress exacerbates physiological and psychological health issues across several different organ systems [7,8]. It has been repeatedly demonstrated that astronauts undergo rapid senescence in space, and disease progression is greatly accelerated relative to on Earth [9]. Fortunately, long-term systematic reconditioning upon return to Earth restores considerable, though not all, spaceflight impairments [10]. However, longer flight durations and potentially even one-way journeys will require novel mechanisms to maintain astronaut health.

Beyond the protection of the low Earth orbit and the Earth’s magnetic field, cosmic radiation is a leading environmental hazard for astronauts. Efficient shielding of high-energy radiation particles from galactic cosmic rays is extremely difficult, and the extent to which radiation exposure would limit long-term space exploration is still uncertain [11]. Radiation exposure can have both acute and long-term consequences, from nausea and decreased lymphocyte count to cancer and degenerative disease development later in life [12,13]. Immune dysregulation can arise [14], potentially resulting from radiation exposure and myelosuppression [15,16], microgravity-induced aberrant bone marrow differentiation [17], disruptions in circadian rhythm [18], or various other stressors experienced during spaceflight. Impairments in immunity limit the ability to fight disease and lead to the loss of regulation of the microbiome. This has manifested in astronauts through the reactivation of latent viruses, skin allergy, and overall hypersensitivity [16,19]. Interestingly, in both laboratory and wild animals exposed to radiation (such as those inhabiting the area of Chernobyl), as well as leukemia patients undergoing radiotherapy, microbiota changes may be potentially protective against the inflammatory effects of the radiation [20,21]. As previously reviewed, bacteria and the microbiome hold immense potential to mitigate radiation injury, immune dysregulation, and decontaminate radionuclides, and targeted microbiome-modulating therapies such as probiotics and FMT continue to gain favor in the literature for these purposes [19,22,23].

In addition to radiation, microgravity is a significant health problem encountered during spaceflight. It is associated with cardiovascular deconditioning, bone and muscle atrophy, kidney stone formation, and numerous other pathophysiological changes [7]. Although it is difficult to tease out the effects of microgravity from the myriad of other environmental challenges experienced during spaceflight, in vitro and in vivo simulated microgravity conditions demonstrate alterations to the microbiota and bacterial cell physiology [24,25]. Specifically, microgravity can enhance virulence in pathogens [3,26] as well as increase the growth rate and environmental stress resistance in commensal microbes [27,28]. *Salmonella typhimurium*, the causative agent of generally self-limiting gastroenteritis, is nearly twice as lethal when cultured in low-orbit microgravity and administered to mice on Earth, compared to ground control cultures [29]. A stark global reprograming in gene expression leads to an increased production of virulence genes that might be responsible for this observation [29]. Beyond the changes to bacterial physiology, microgravity can also lead to decreased intestinal motility [30], altered gastric secretions, and increased intestinal permeability, all of which are intimately connected to the gut microbiome’s composition and overall gastrointestinal health [31,32]. Studies in mice have suggested that the gut microbiota significantly shifts during both true spaceflight and simulated microgravity, and these shifts are coupled with innate immune suppression [24,33]. Interestingly, true spaceflight shows additional alterations to the microbiota beyond what is observed in the ground control animals, which were diet matched and exposed to a simulated microgravity environment, illustrating unique attributes of the bona fide spaceflight experience that manifest in microbiome shifts [33,34].

Astronauts also encounter numerous psychological alterations as a result of space travel. Motion sickness, claustrophobia, depression, anxiety, and changes in circadian rhythm are commonly experienced by members of the flight crew, and these concerns make them more prone to infection, inflammation, and on-board errors [35,36]. These conditions could be further compounded by resultant influences on the microbiome [37]. Alterations to circadian rhythm such as those experienced by astronauts may influence the gut microbiota, because bacterial community composition oscillates alongside host-linked diurnal fluctuations [38,39]. Furthermore, the perturbation of the rhythmic microbiota through insomnia, altered eating patterns, and other disruptions could affect neurochemical signaling via the gut–brain axis, which in turn may exacerbate mental illness [37]. These trends all mirror results in astronauts whose microbiota and immune function were altered following spaceflight, although the contribution of microgravity vs. radiation vs. the plethora of confounding environmental and human factors remains to be determined [1,40]. Regardless, it is clear that the intimate role the microbiome plays in overall health extends to spaceflight and its pathophysiological adaptive changes (Figure 2).

## 3. The Healthy Gut Microbiota Is an Essential Provisioner

Unlike other organ systems, the “anatomy” of the human microbiome is radically variable between individuals. While many studies report generalized “dysbiosis” in disease states, this is relatively uninformative since there is no singular healthy or “eubiotic” microbiota [41,42]. Rather, a healthy microbiota is environmentally and genetically contextual, with recent literature from large cohort studies citing that diverse factors and still elusive effects predominantly drive microbiota variation between individuals [43,44,45]. Although there is no one-size-fits-all approach to assessing the health of an individual’s microbiota, a few attributes are consistent, including a high degree of diversity and low pathobiont burden. While many gut microbiota functions are conserved across diverse groups of people, taxonomically the same bacteria are not always detected [46]. In the gut, high microbial diversity is associated with health, and a loss of diversity is often observed with aging and disease development [46].

In comparison to infection-causing pathogens, evolutionarily well-adapted symbionts often demonstrate genetic minimalism and auxotrophy, necessitating extracellular cooperative sharing of micronutrients [41,47,48]. Recent evidence suggests that well-adapted microbiomes develop an interconnected electron transport chain, with key resources including aromatic amino acids, metabolic cofactors (B vitamins), menaquinones (vitamin K2), hemes, and short-chain fatty acids being shared in the extracellular space [41]. This benefits not only the other microbes in the resource-sharing network, but also the host, because the microbiota produces these constituents at biologically meaningful concentrations for host utilization [41,49]. Importantly, some of the micronutrients known to be produced or modified by the gut microbiota may be deficient in astronauts during spaceflight [50,51].

Although not previously thought to have a microbiome connection, the bone loss that occurs as a result of microgravity in space may be bacterially influenced, as it is associated with calcium, vitamin D, and vitamin K deficiency [52]. Vitamin D regulates multiple biological processes, including the absorption, excretion, and storage of calcium. It is normally obtained in the diet, through supplements, and via ultraviolet exposure from the sun. However, astronauts are heavily shielded, thus preventing endogenous UV-induced synthesis and resulting in serum vitamin D deficiency that persists despite daily supplementation during spaceflight [51,53]. Gut bacteria may influence vitamin D homeostasis by converting between vitamer forms [54], degrading vitamin D [55], or altering its bioavailability and activity [56,57]. In fact, differences in the blood profiles of vitamin D vitamers are associated with shifts in the gut microbiota [58,59]. Vitamin K is involved in calcium homeostasis through the carboxylation of osteocalcin and matrix Gla protein, which aid in the deposition of calcium in bones and away from soft tissues, respectively [60]. While vitamin K1 (phylloquinone) is plant derived, vitamin K2 (menaquinone) is produced by the gut microbiota. This K2 improves bone mineral density [61], but supplementation through oral intake of both K2 and D in the elderly has been shown to be ineffective at ameliorating aortic valve calcification [62]. This may also be the case in astronauts who have been in space for 6 months or more who develop arterial stiffness during flight [63], while other astronauts present improvements in bone metabolism markers upon vitamin K supplementation [50]. The theory has been that during spaceflight, calcium from bones ends up being resorbed into systemic circulation where it can travel to other places, whether that be in the arteries or kidneys where it can form mineral deposits and stones [64], but this notion is probably an oversimplification. For example, while many resources cite calcium-based stones as the predominant composition in astronauts, a 2008 report concluded that 60% of stones analyzed were of unknown constituents [65]. Thus, in these poorly understood spaceflight health outcomes, a more complex mechanism is likely at play, which involves the gut microbiome’s modulation of vitamins and other bioactive compounds.

## 4. Key Microbes?

Voorhies et al. (2019) demonstrated that the gut microbiota of crew members converges during spaceflight [1]. There was also a reduction in the abundance of key symbionts including *Bifidobacterium* and *Akkermansia* spp., indicating an environment-wide decrease in diversity, despite intraindividual diversity often remaining unchanged for 6–12 months. With no method in place to curtail this loss of diversity, and a lack of environmental microbial exposures or bacterial replenishment aboard a spacecraft, astronaut health may be compromised.

Microbes are critical to the survival of humans and many other life forms. For example, in *Drosophila melanogaster*, the deprivation of exposure to new microbes leads to a shorter lifespan [66,67]. Our own evolution suggests that the cecal appendix, once considered useless, may in fact serve an evolutionarily adapted purpose through its association with longevity [68]. Indeed, research now points to its role as a reservoir of the gut microbiota, able to re-seed after disruptive episodes. However, if key microbial species are already “missing” or low in abundance, it is unknown whether this re-seeding would effectively improve health.

Most probiotic supplements are composed of a short list of species, often from dairy or gastrointestinal origins, and exert transient beneficial effects without persistence or colonization. This does not ensure microbial diversity is restored unless their metabolic activity enhances the growth of other gut organisms. So, while probiotics provide remarkable benefits for numerous health conditions, regular exposure to more diverse microbial sources may be preferred for health maintenance (Figure 1). Recent studies have demonstrated that the average American diet, which is high in processed foods, contains very few live microbes [69,70]. Wastyk et al. (2021) documented the benefit of a gut microbiota-targeted diet (incorporating fermented foods or prebiotics) in stimulating indigenous commensals, altering microbiota function, and augmenting diversity [71]. However, dietary intake is not the only source of microbial exposure that is depleted in Western society. The hygiene hypothesis has suggested that reduced microbial exposure impairs immune development leading to allergies, and research conducted in the years since then, both on Earth and in space, has corroborated these ideas [72].

Individuals who are routinely exposed to large numbers of diverse microbes, such as those who live in rural environments, have lower rates of asthma, allergy, inflammatory, and even psychiatric disorders [73,74]. Perhaps one of the most compelling studies of lifestyle on the microbiota was conducted in the nomadic traveler communities of Ireland [75]. Irish travelers are genetically similar to the settled Irish population, but through recent policy decisions, some of their communities have become sedentary in established permanent housing. In the study population, this forced modernization resulted in a drastic lifestyle change compared to their childhood nomadic lifestyle, with large families in close living quarters and in proximity to animals. The microbiota between the settled and unsettled groups was significantly different; specifically, the microbiota in the sedentary group resembled that of industrialized Western society and was linked to metabolic disease risk factors. This correlates with the sharing of astronaut microbiotas living in the International Space Station (ISS), and again emphasizes the need to maintain and restore microbial diversity, both within and between astronauts. Of course, this does not guarantee that astronaut health would improve, but the idea is worth considering.

Our health is dependent upon that of our microbial exposome, but the concept that we are part of a larger ecosystem as a “holobiome” (the host and species around it) is hard for some to consider [41]. The microbiota of the spaceship itself is not likely to vary much, as studies on the ISS have demonstrated [76]. The predominant organisms on the ISS were those that are associated with humans, as well as extremophiles, which were found in less hospitable areas of the station. The historical case of Legionnaire’s disease still comes to mind in such systems, where seemingly inert environmental microbes can become a serious threat when delivery vectors and depleted immunity present themselves [77]. In the future, a more effective strategy to foster astronaut health while mitigating infectious threats may be to optimize the environmental microbiota in the spacecraft prior to departure, rather than just attempting to eliminate it [78]. Sanitation of inanimate objects and surfaces with competing, but not harmful, microbes such as *Bacillus* sp. is an emerging tactic for reducing hospital-acquired infections [78].

In addition, some microbiota–host genetic screening and preselection criteria could be implemented to determine astronaut candidates who may be more resistant to microbiota changes that adversely impact their health. Such outlier individuals have been observed previously, and they may thrive during long-term spaceflight [1]. The concept of taking a “Noah’s Ark” of microbiota into space may not be so far-fetched as some might believe.

## 5. Solutions

The use of probiotics, prebiotics, and selected other microbiota-modulating therapies using existing products for potential benefit in spaceflight has been thoroughly summarized by others [8,79,80,80]. Monoculture probiotics as well as those containing multiple strains, while a long way from an ecosystem replacement, have shown some remarkable health benefits, potentially warranting their regular use in space. As previously discussed, spaceflight can reactivate latent viruses, allergy, and overall hypersensitivity, which on Earth have all been shown to be mitigated with probiotics [82]. For example, orally consumed probiotic bacteria have demonstrated the ability to lower Epstein–Barr (EBV) and cytomegalovirus (CMV) antibody titers [83], which is particularly relevant given the known reactivation of EBV, CMV, and varicella-zoster virus in astronauts experiencing long-duration spaceflight [84]. Beyond the robust data from orally consumed probiotics, it is possible that nasally administered or inhaled probiotics may provide additional clinical benefit to astronauts. Specifically, when the immunomodulatory properties of probiotics are the outcome of interest, nasally inhaled bacteria make immediate contact with the mucosal nasopharyngeal tissues, in contrast to the potentially hours-long period after gastric consumption [85]. These delivery methods have shown divergent and beneficial immune responses in mice when recombinant antigen-specific lactic acid bacteria (LAB), as well as purified antigens administered alongside LAB, were tested against both dietary and airborne inhalant allergens [86,87,88]. Topical probiotics have also demonstrated promise in treating dermatitis, rosacea, and skin wounds [89], all of which have been documented to occur during spaceflight [16]. Thus, not only oral, but also inhaled, topical, and even novel delivery methods of probiotics may be worth considering for administration to astronauts for the targeted treatment of specific medical conditions, or regular microbial exposure and microbiota maintenance.

As discussed in Section 2, environmental hazards such as microgravity can greatly affect bacterial physiology, and since deep space exploration will involve trips that are years in length, the shelf life of probiotics poses a problem, though not insurmountable [90]. FMTs, whether autologous or allogeneic (for example, from an astronaut’s own pre-spaceflight fecal sample), have the potential to replenish crew microbiota as the composition changes and diversity diminishes in space; however, as is the case with probiotics, the viable components decline over time. Thus, these tactics may not offer the most efficacious long-term results and innovative new interventions specific to spaceflight should be considered.

Freeze-dried and other preserved foods may suffice nutritionally during shorter spaceflight, but over longer periods, nutritional deficits often develop [91]. Spaceships instead may need to become living vehicles or “earthships” to recycle carbon and nitrogen within the system by farming to sustain human health indefinitely. Biosphere projects on Earth, essentially enclosing people with sustaining flora and fauna within finite space, have not always been successful and require miniaturization, but have demonstrated success in achieving food self-sufficiency [92]. Incorporating the lessons learned from these initial investigations, traveling for years in space will require additional approaches to ensure dietary needs are met, such as growing grains, fruits, vegetables, fungi, insects, and potentially even “lab meat” cultivated from cells. While processed foods are shelf-stable, the vitamins and essential micronutrients in pre-packaged meals are still labile and would degrade during long space travel [93]. Furthermore, packaging adds weight and requires restocking, whereas if carbon and nitrogen were recycled through natural growth systems, this may reduce the dependence on packaged, microbially deplete foods. Insects, microalgae, and mushrooms could be particularly high-value space-grown “crops”, as they are nutritionally and microbially dense, and can grow on non-edible biomass or sewage with relatively little husbandry [94]. Mushrooms specifically are high in vitamins, minerals, protein, oligosaccharides, and other prebiotic fibers to benefit the astronaut as well as their microbiome [95]. Additionally, growing mushrooms can regenerate soil [96], be used to make batteries [97], and recent research at NASA has even looked at the use of mushrooms to grow “myco-architecture” and other structures in extra-terrestrial environments [98].

To further attend to the health of the astronaut microbiome, additional nutritional and immunological benefit may be obtained by the fermentation of these freshly grown products [99]. Fermentation of food en route would provide a degree of preservation while also improving palatability and dietary variation, a valuable contribution as menu fatigue leads to decreased food consumption and the development of caloric and micronutrient deficiencies [91]. This process can also remove potent antinutrients including phytates and oxalates, which can further deplete an astronaut’s micronutrient levels and pose health risks [100,101]. Though not directly food-related, such a process could also be utilized to produce resources such as ethanol and other industrial compounds, either as a target or by-product. Again, these multi-functional products could offer numerous health and efficiency benefits over sterilized and pre-processed foods shipped from Earth.

The variety of fermentable foods is lengthy, adding to sensory enjoyment and contributing to mental health [102,103]. Organisms used to ferment unique products in different parts of the world (which may have not been adopted by other cultures due to their unique organoleptic and texture properties) may harbor tremendous potential. For example, natto, a Japanese soy product fermented by *Bacillus* species is a potent source of vitamin K2, with proven benefits toward bone health [104,105]. These *Bacillus* sp. may also hold promise as potential probiotics in mitigating calcium oxalate kidney stones [106], which are a significant threat to long-term space travel [65]. In another study by Hati et al. [107], *Lactobacillus* isolates from traditional fermented Indian foods demonstrated potent B vitamin and short-chain fatty acid production, both of which can become dysregulated in astronauts [108] and the simulated space environment [109]. We are yet to ferment foods from exciting new ingredients such as laboratory-created meats or milk, or using genetically engineered microbes with benefits such as enhanced vitamin or nutrient outputs [110]. While considered relatively basic science, ideal food and microbe matches to produce tasty and nutritious fermented foods efficiently need to be identified for spaceflight utilization. Rather than the sterilized space food of the past, these innovative food systems will contribute to bacterial exposure for the astronauts in not only the food’s consumption, but also environmentally through its growth and husbandry.

## 6. Monitoring and Manipulating the Microbiota for Health

As discussed earlier, it appears that what constitutes a healthy microbiome is dependent on its function, rather than its taxonomic composition. We should therefore be evaluating the microbiota with regard to its functional contribution to the holobiome. As synthetic biology advances, reporter microbes may be important in monitoring microbiome health during future spaceflight. These biosensors could signal the collapse of bioenergetic nodes, a bloom of pathobionts, and other undesirable metabolomic outputs [111]. Additional developments utilizing CRISPR-Cas tools may enable simultaneous monitoring of multiple biomarkers and could be employed to engineer solutions to the detected problem [112]. Future space teams may be armed with gene-editing capabilities to remedy unforeseen health challenges in real time as they arise. These may include a loss of specific microbes or microbial functions, or a gain of pathogenic ones. There is strong evidence that the microbiota influences the efficacy of many chemical and immune therapeutic options [113,114]. FMTs are now being investigated to transfer the microbes responsible for a desired immune or xenobiotic response, though this may not be the best option for space travel. Rather than sending huge microbial biobanks into space, perhaps we could simply provide specific workhorse microbes capable of genetic editing to produce versatile bioactive metabolites and shift the microbiome and disease phenotype.

## 7. Conclusions

Future space travelers will face similar issues to those of the early explorers of Earth, but with even greater challenges. Early maritime travelers mitigated nutritional woes by providing fresh and fermented foods to protect against vitamin deficits (e.g., scurvy), and technical innovations were necessitated to preserve foods with a longer shelf life, thus enabling bolder geographic exploration. Similarly, many practical innovations will need to be met before long-duration space travelers can become self-sufficient. The microbiota and its replenishment through environmental and dietary exposures are essential for our continued health, particularly in adverse conditions. Although probiotics of various types have demonstrated efficacy for many afflictions that could occur during space travel, further consideration needs to be given to not just the supplementation of specific microbes, but also the maintenance of the holobiome. With the incorporation of freshly grown produce, new foods such as lab meat, and fermented foods into the space diet, in addition to synthetic biological tools and novel tactics to modulate the environmental exposome, we can work to optimize the microbiome for healthy and successful long-term space travel.

## Figures and Tables

**Figure 1 life-12-01163-f001:**
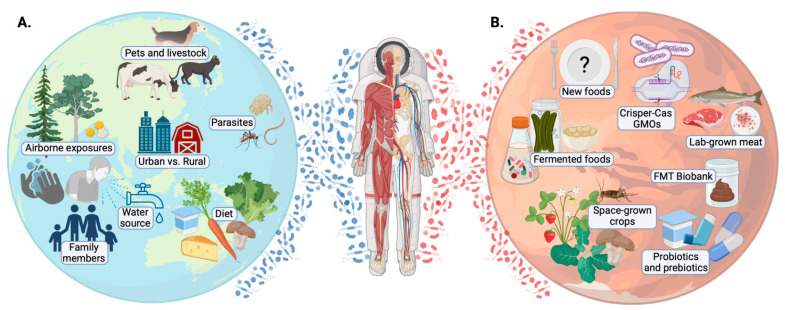
The health of humans and their microbiomes are influenced by the microbes they encounter in their environment, fermented foods, probiotics, and part of their standard diet. (**A**) On Earth, the microbes that come from the animals, plants, and the wider biosphere are important to our health. (**B**) Long-duration spaceflight or life on Mars in isolation will require microbial enhancement through different strategies. These may include: dosing with environmental microbes; establishing a complex biosphere of space-grown crops; consuming fermented foods, probiotics, or prebiotics; collecting healthy samples prior to space departure to allow reintroduction by FMT when required at a later stage; and use of genetic modification tools. Created with BioRender.com.

**Figure 2 life-12-01163-f002:**
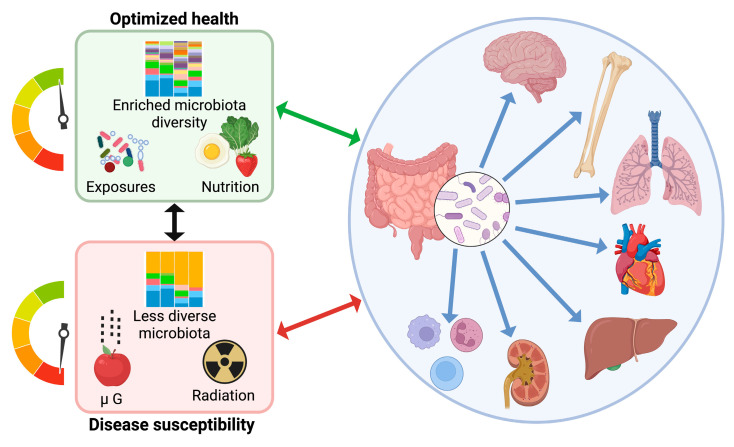
Factors in space such as microgravity (µG) and radiation place considerable stress on different human systems (mental, musculoskeletal, cardiovascular, hepatic, urological, immune, etc.). The healthy microbiota assists in our ability to deal with these stresses, but if the microbiota changes adversely over time due to isolation and nutritional deficits our risk of disease increases. Created with BioRender.com.

## Data Availability

Not applicable.

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
