# Peer review of "Long-Duration Space Travel Support Must Consider Wider Influences to Conserve Microbiota Composition and Function"

_life, 2022, doi:10.3390/life12081163_

Round 1

Reviewer 1 Report

The manuscript is well written and the exposure of the contents is very clear. I only have a couple of comments before accepting it for publication:

- please add the appropriate reference numbers next to the names of the authors cited directly, on lines 183 and 205.

- shelf life of probiotics and food sure represents a problem, but does it depend only on the time elapsed between preparation and consumption, or also on the adverse conditions of the spaceflight, as described in the second paragraph?

Author Response

Thank you for your assessment of the manuscript.

- We have amended the references to reflect the reviewer's suggestion.

- We agree with the reviewer that time as well as factors unique to spaceflight such as microgravity have the potential to affect shelf life (of not only food and probiotics, but also FMT and any other live biotherapeutic agent brought from Earth). In the second paragraph of section 5, we have amended the sentence for clarity. We now state “As discussed in section 2, environmental hazards such as microgravity can greatly affect bacterial physiology, and since deep space exploration will involve trips that are years in length, the shelf life of probiotics poses a problem, though not insurmountable”.

Reviewer 2 Report

The review drafted by Kait et al. focuses on factors that need to be considered to maintain a healthy microbiota in astronauts during long space travel. This review is appropriately designed to include essential sections and informative to a broader audience to appreciate the challenges in maintaining normal microbiota. This review also has well informative figures, some of which can be improved. Below are the reviewer’s critiques.  

The title can be re-worded to reflect the review content. For example, ‘Long-duration space travel support needs consideration of many factors to preserve structure and function of microbiota’

Figure 1. This figure can be made more informative to the reader. On the left side image, replace the graph with a reduced microbial diversity graph similar to the one on the right side of the image. Use connecting arrows in both images to connect it to microbial diversity. The top of the image indicates red color as a disease state while green healthy state. Spell out uG as microgravity in the legend section.

Add a new figure indicating the importance of the gut microbiome in different human axis systems. Gut-brain, Gut-lung, Gut-liver, etc.  Connect these axes to a commonly observed negative effect on astronauts. The message of the figure should be, that by preserving gut microbiota, other organs’ functions can be well-maintained. Connect this figure to 5. Solutions section.  

Author Response

Thank you for your assessment of the manuscript.

- We have revised the title to more accurately reflect the content of the review. "Long duration space travel support must consider wider influences to conserve microbiota composition and function"

- We have amended Figure 1 to include the reviewer’s suggestions, as well as their valuable points about an additional figure.